# Feasibility of thin-slice abdominal CT in overweight patients using a vendor neutral image-based denoising algorithm: Assessment of image noise, contrast, and quality

**Akio Tamura**[1]*, **Manabu Nakayama**[1], **Yoshitaka Ota**[2], **Masayoshi Kamata**[2], **Yasuyuki Hirota**[2], **Misato Sone**[1], **Makoto Hamano**[1], **Ryoichi Tanaka**[3], **Kunihiro Yoshioka**[1]

1 Department of Radiology, Iwate Medical University School of Medicine, Morioka, Japan, 2 Division of Central Radiology, Iwate Medical University Hospital, Morioka, Japan, 3 Division of Dental Radiology, Department of General Dentistry, Iwate Medical University School of Dentistry, Morioka, Japan

* a.akahane@gmail.com

## Abstract

The purpose of this study was to investigate whether the novel image-based noise reduction software (NRS) improves image quality, and to assess the feasibility of using this software in combination with hybrid iterative reconstruction (IR) in image quality on thin-slice abdominal CT. In this retrospective study, 54 patients who underwent dynamic liver CT between April and July 2017 and had a body mass index higher than 25 kg/m$^2$ were included. Three image sets of each patient were reconstructed as follows: hybrid IR images with 1-mm slice thickness (group A), hybrid IR images with 5-mm slice thickness (group B), and hybrid IR images with 1-mm slice thickness denoised using NRS (group C). The mean image noise and contrast-to-noise ratio relative to the muscle of the aorta and liver were assessed. Subjective image quality was evaluated by two radiologists for sharpness, noise, contrast, and overall quality using 5-point scales. The mean image noise was significantly lower in group C than in group A (p < 0.01), but no significant difference was observed between groups B and C. The contrast-to-noise ratio was significantly higher in group C than in group A (p < 0.01 and p = 0.01, respectively). Subjective image quality was also significantly higher in group C than in group A (p < 0.01), in terms of noise and overall quality, but not in terms of sharpness and contrast (p = 0.65 and 0.07, respectively). The contrast of images in group C was greater than that in group A, but this difference was not significant. Compared with hybrid IR alone, the novel NRS combined with a hybrid IR could result in significant noise reduction without sacrificing image quality on CT. This combined approach will likely be particularly useful for thin-slice abdominal CT examinations of overweight patients.

**Data Availability Statement:** All relevant data are within the paper and its Supporting Information

files. Additional data are available upon request to the corresponding author.

**Funding:** This work received funding from JSPS KAKENHI Grant Number 19K08160 to AT. The funder had no role in study design, data collection and analysis, decision to publish, or preparation of the manuscript.

**Competing interests:** The authors have declared that no competing interests exist.

## Introduction

Computed tomography (CT) plays an important role in diagnosis and therapeutic management. The use of CT has increased drastically with technical developments and new applications, as has the associated radiation exposure of the population [1]. Several hybrid iterative reconstruction (IR) methods, such as adaptive statistical IR (ASiR, GE Healthcare), iDose (Philips Healthcare), IR in image space (IRIS, Siemens Healthcare), sinogram-affirmed IR (SAFIRE, Siemens Healthcare), and adaptive iterative dose reduction (AIDR, Canon medical systems), have been proposed for reducing the radiation dose by decreasing image noise during the reconstruction process. Currently, hybrid IR is a popular choice for reducing image noise and improving image quality.

Recent advancements in MDCT have allowed image acquisition with thinner collimation and more rapid scan times, thus, enabling better image resolution to delineate abdominal disease and abnormalities. In most clinical practice guidelines and recommendations for patients with abdominal cancers, such as pancreatic cancer and cholangiocarcinoma, the CT technique involves multiphase thin-section image acquisition to optimize the evaluation of spread to the blood vessels [2–4]. The thinner layer provides better detail and spatial resolution; conversely, noise in CT image increases with a thinner slice [5]. In addition, abdominal CT examinations in overweight patients remain challenging owing to the substantially higher radiation doses required and are associated with a substantially decreased low-contrast detectability compared with examinations in non-overweight patients [6]. For abdominal CT examinations, high image noise levels are a critical issue because the noise may obscure subtle low-contrast lesions in parenchymal organs [7,8]. CT image noise strongly depends on the patient's body size and the tube current applied during data acquisition. It is theoretically a priori possible to increase the number of photons and, thus, increase the radiation dose, and to obtain image quality even with a 1-mm slice compared to a 5-mm slice acquired with a lower tube current [8].

Recently, a next-generation noise reduction algorithm, model-based IR (MBIR), a more advanced IR technique compared with hybrid IR, was introduced. Results of studies have shown that MBIR leads to substantial dose reductions but maintains diagnostic image quality and reduces image noise [9–11]. However, it may be cost prohibitive as replacement of the CT scanner may be required for using MBIR. Another option is third-party image-based denoising. These image-based methods cost much less and can be useful in CT practices that employ multiple scanner vendors and models. A recently developed new image-based noise reduction algorithm including SafeCT and iNoir, a third-party vendor neutral imaging-based approach, which is not linked to the image reconstruction system of a CT scanner but works on the workstation, can be applied to DICOM data to remove noise. However, data on the impact of the third-party image-based denoising algorithm for abdominal CT imaging are limited.

Thus, the purpose of this study was to investigate whether the novel image-based noise reduction software (NRS) improves image quality and to assess the feasibility of using this software in combination with hybrid IR in image quality on thin-slice abdominal CT for moderately overweight patients.

## Materials and methods

### Patient selection

This retrospective study was approved by institutional review board of Iwate Medical University, and the requirement for written informed consent was waived because the image data were retrospectively obtained from routine liver CT examinations. Patients who underwent dynamic liver CT between April and July 2017 were evaluated. Patients were included in the

study if they had a body mass index (BMI) higher than 25 kg/m$^2$ and if they were aged 20 years or older. Exclusion criteria were contraindications to iodinated contrast media, emergency cases, and hemodialysis or renal failure. Therefore, 54 patients (45 men, 9 women; age range, 28–84 years; mean age, 65.0 years; body weight [BW] range, 70–112 kg; mean BW, 78.3 kg; height range, 150–184 cm; mean height, 166.3 cm; BMI range, 25.0–39.3 kg/m$^2$; mean BMI, 28.4 kg/m$^2$) were included in our study.

## CT technique

All examinations were performed on an MDCT scanner (Aquilion 64, Canon medical systems, Ohtawara, Tochigi, Japan) using the following parameters: 1-mm section thickness and interval, 0.5 s rotation time (fixed), 120 kVp tube voltage, and under automatic exposure control with a noise index of 10. Nonionic contrast medium containing an iodine concentration of 300 mg/mL (Omnipaque 300; Daiichi Sankyo, Tokyo, Japan) at a dose of 2 mL per kg of BW was injected into the antecubital vein using a power injector (Nemoto DUAL SHOT Type D, Tokyo, Japan) with a fixed injection duration of 30 s. The scan delay for arterial and portal venous phase imaging were determined using an automatic bolus-tracking program (Canon medical systems). Scanning for the arterial and portal venous phases was started automatically 20 and 60 s, respectively, after the trigger threshold (100 HU) was reached at the level immediately above the celiac trunk. Pre-contrast images and equilibrium phase images were also obtained for all patients; for the equilibrium phases, a fixed-time delay of 180 s was applied. However, pre-contrast and equilibrium phase images were not evaluated in this study.

## CT image reconstruction

The raw data scanned in the arterial and portal phases were reconstructed with hybrid IR (AIDR 3D mild). These images were reconstructed with 1-mm slice thickness and 5-mm slice thickness. The images reconstructed with 1-mm slice thickness were also denoised using a commercially available workstation (Virtual Place iNoir; AZE, Kanagawa, Japan). We selected an NRS algorithm level of 75% as recommended by the vendor for abdominal imaging. In summary, three image sets of each patient were reconstructed as follows: hybrid IR images with 1-mm slice thickness (group A), hybrid IR images with 5-mm slice thickness (group B), and hybrid IR images with 1-mm slice thickness denoised using NRS (group C).

## Objective assessment of image quality

A single CT technologist (M.K., 30-year experience in body CT) measured the mean attenuation of the abdominal aorta, hepatic parenchyma, and bilateral erector spinae muscles with circular regions of interest (ROIs) of the three image sets. Attempts were made to maintain a ROI area of 150 ± 25 mm$^2$. For all measurements, the size and position of the ROIs were maintained constant among the three image sets by applying a copy-and-paste function at the workstation. Aortic attenuation was measured immediately above the level of the celiac trunk in the arterial phases. Hepatic attenuation was measured in three separate areas (left lobe and the anterior and posterior segments of the right lobe) on images obtained at the level of the main portal vein in the portal venous phases (Fig 1). Areas of focal changes in parenchymal density, large vessels, and prominent artifacts were carefully avoided. Attenuation of the two erector spinae muscles was measured in the arterial and portal venous phases without including macroscopic areas of fat infiltration. To ensure consistency, all measurements were performed three times and mean values were calculated. Image noise was defined as the standard deviation (SD) of the attenuation value measured in the erector spinae muscle. The contrast-to-noise ratio (CNR) of the abdominal aorta was calculated by subtracting the ROI of the aorta

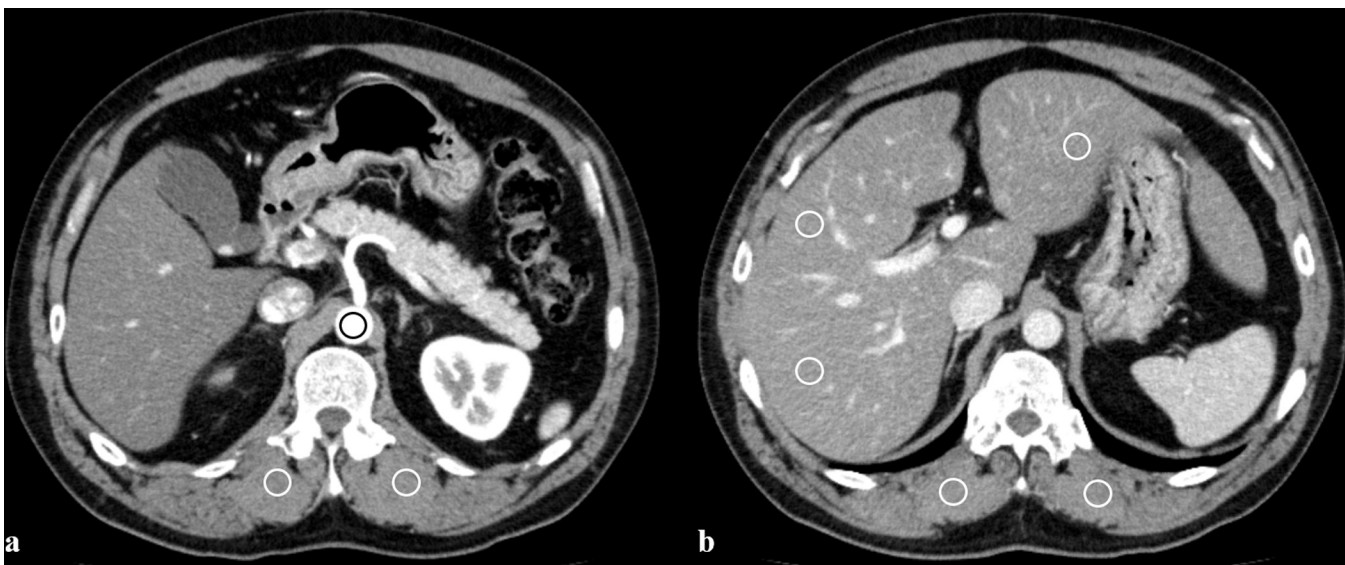

**Fig 1.** Axial contrast-enhanced CT images obtained in a 49-year-old man during the arterial phase (a) show regions of interest (ROIs) manually drawn on aorta and bilateral erector spinae muscles; the portal phase (b) shows ROI drawn on liver and bilateral erector spinae muscles. For all measurements, the size and position of the ROIs were maintained constant among the three image sets by applying a copy-and-paste function at the workstation.

from the ROI of the erector spinae muscle and then dividing this difference by the image noise. The CNR of the hepatic parenchyma was calculated by subtracting the ROI of the liver from the ROI of the erector spinae muscle and then dividing this difference by the image noise.

## Subjective assessment of image quality

Two radiologists (M.N. and A.T., with 8 and 9 years of experience in abdominal CT, respectively) independently graded the images for sharpness, contrast, noise, and overall quality. The CT data sets were randomized, and the name, age, and sex of the patients, as well as the CT parameters and all hospital record numbers, were removed from the images. CT images were presented with a standard abdominal window (window width, 350 HU; window level, 40 HU). The radiologists used portal phase axial CT images to evaluate the 1-mm slice thickness (group A), 5-mm slice thickness (group B), and 1-mm slice thickness denoised using NRS (group C). The readers used a five-point subjective scale to grade image sharpness (1 = blurry, 2 = poorer than average, 3 = average, 4 = better than average, and 5 = sharpest). Image noise was graded as follows: 1 = unacceptable noise, 2 = above-average noise, 3 = average noise in an acceptable image, 4 = less-than-average noise, and 5 = minimum or no noise. Image contrast and overall image quality were graded as follows: 1 = unacceptable, 2 = suboptimal, 3 = average, 4 = above average, and 5 = excellent.

## Statistical analysis

All numeric values are reported as mean ± standard deviation. To compare quantitative and qualitative image analyses among groups A, B, and C, we used a multiple comparison test (Dunnett's test). P values lower than 0.05 were considered to indicate statistical significance. Interobserver agreement was measured with kappa statistics. The scale for kappa coefficients for interobserver agreement was as follows: less than 0.20 = poor, 0.21–0.40 = fair, 0.41–0.60 = moderate, 0.61–0.80 = substantial, and 0.81–1.00 = near-perfect. Statistical analysis was

performed with SPSS (version 24, IBM Corp., Armonk, NY, USA) and R (version 3.4, R Project for Statistical Computing, Vienna, Austria).

### Phantom experiment

In addition to the patient examinations, we performed phantom experiments to evaluate the noise power spectrum (NPS) and modulation transfer function (MTF). The noise image data were acquired from the 220-mm diameter cylindrical water phantom at the 120 kVp/300 mA setting, with 32×1-mm collimation, with and without hybrid IR. The three image sets—images reconstructed with FBP, images reconstructed with FBP using NRS, and images reconstructed with hybrid IR—were acquired for measuring NPS and MTF (Fig 2). By placing an ROI of 256 × 256 pixels at the center of the image (Fig 2E), the NPS was calculated by the radial frequency method using the CT measure version 0.97b (Japanese Society of CT Technology, Hiroshima, Japan). To improve the accuracy and account for statistical uncertainties of the NPS data, 50 scans were performed with the same table position for each protocol, and a total of 50 NPS curves were averaged for each protocol. For MTF analysis, the 220-mm diameter cylindrical water phantom containing nylon (100 HU) and delrin (340 HU) was used. To acquire the MTF analysis, 50 scans were performed with the same table position for each protocol, and a total of 50 MTF curves were averaged for each protocol. For each averaged image, an ROI was placed around the two objects as shown in Fig 2F. MTF values were calculated using the radial edge method, with contrasts of 100 and 340 HU using the CT measure version 0.97b [12].

## Results

### Objective assessment of image quality

The mean image noises for the erector spinae muscles on the arterial and portal venous phase images were significantly lower in group C than in group A ($p < 0.01$, Table 1), but no significant difference was observed between groups B and C ($p = 0.98$ and 0.86, respectively). The CNR for the abdominal aorta and hepatic parenchyma was significantly higher in group C than in group A ($p < 0.01$ and $p = 0.01$, respectively). However, there were no significant differences in the CNR between groups B and C (abdominal aorta: $p = 0.82$, hepatic parenchyma: $p = 0.62$).

### Subjective assessment of image quality

Table 2 shows the results of the qualitative analysis. There were significant differences in image noise and overall image quality between groups A and C ($p < 0.01$), but not in sharpness ($p = 0.65$). The contrast of images in group C was greater than that in group A, but this difference was not significant. There were significant differences in sharpness, image noise, contrast, and overall image quality between groups B and C ($p < 0.01$, $p < 0.01$, $p = 0.03$, and $p = 0.01$ respectively), but image noise was graded higher in group B ($p < 0.01$). There was substantial to near-perfect interobserver agreement with respect to sharpness, image noise, contrast, and overall image quality ($k = 0.70$, 0.84, 0.83, and 0.77 respectively). A representative case is shown in Fig 3.

### Phantom experiment

The results of the NPS measurements are presented in Fig 4A. Compared with the image sets reconstructed with FBP and with hybrid IR, the image sets reconstructed with FBP and denoised using the NRS algorithm yielded quantifiable noise reduction across the entire

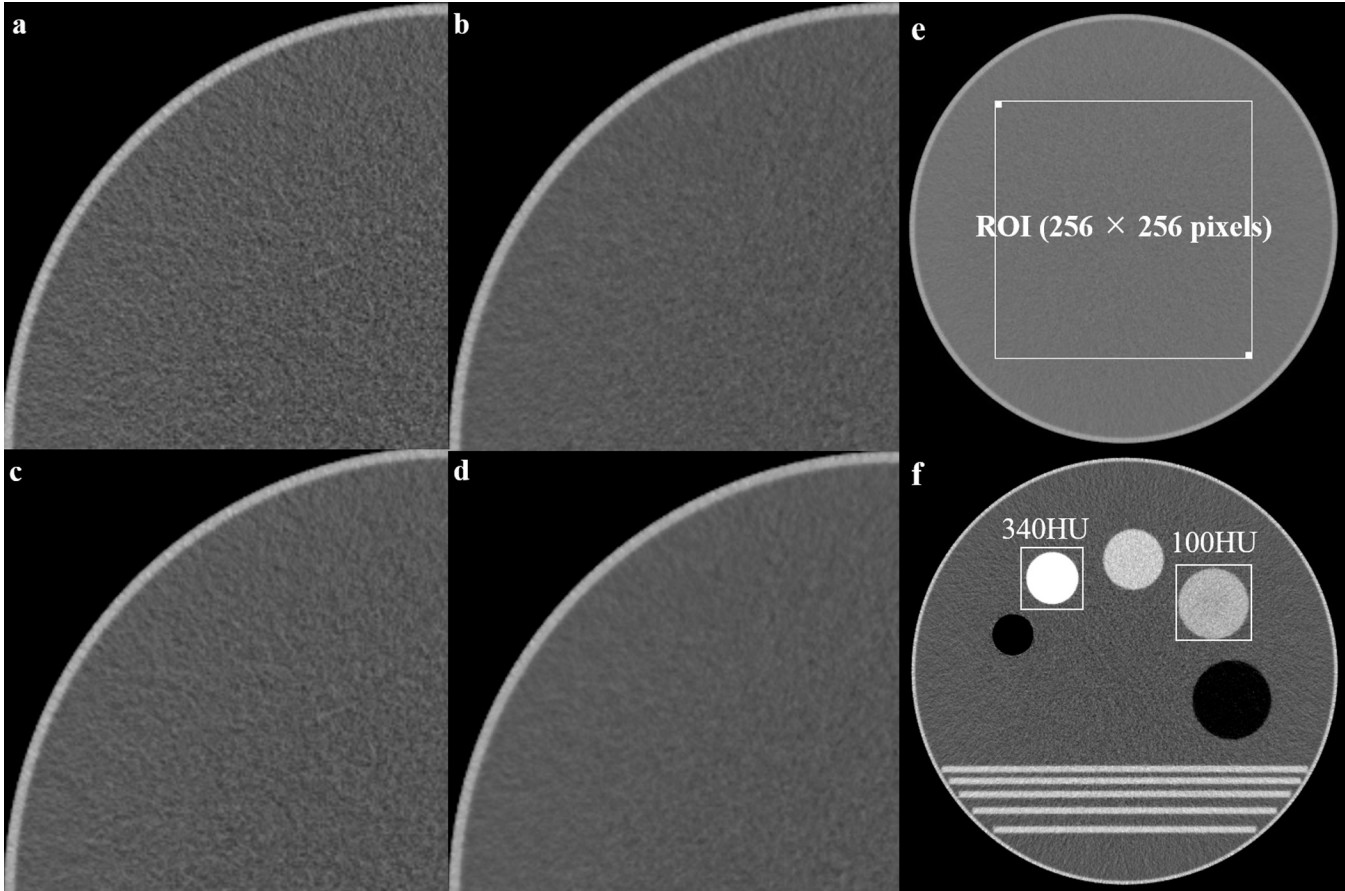

**Fig 2.** CT images show the 220-mm diameter cylindrical water phantom at the 120 kVp/300 mA (a) FBP, (b) FBP using NRS, (c) hybrid IR, and (d) hybrid IR using NRS. (e) Phantom image used for the noise power spectrum (NPS) analysis. The NPS was calculated using the radial frequency method from a region of interest (ROI) and used to calculate the relative noise and peak frequency for each protocol. (f) Image used for the modulation transfer function (MTF) analysis. The MTF values were calculated using the radial edge method, from three ROIs, at contrasts of 100 and 340 HU for each protocol.

spectrum of spatial frequencies. In particular, NRS shifted the NPS toward lower frequencies than those observed with hybrid IR. When the combination of images reconstructed with hybrid IR and with the NRS were evaluated, similar shift and diminishing noise magnitude were found. In contrast, the four curves are very similar in shape and position at 340 HU, the

**Table 1. Objective assessment of image quality in the three image sets.**

| | | | | *p value | |
|---|---|---|---|---|---|
| Parameter | Group A | Group B | Group C | Group A vs Group C | Group B vs Group C |
| Contrast-to-noise ratio | | | | | |
| Aorta | 19.1 ± 6.1 | 28.0 ± 10.0 | 27.1 ± 9.3 | < 0.01 | 0.82 |
| Hepatic parenchyma | 3.4 ± 1.4 | 4.9 ± 2.1 | 4.6 ± 1.9 | 0.01 | 0.62 |
| Noise | | | | | |
| Arterial phase | 17.8 ± 4.5 | 13.1 ± 5.6 | 13.3 ± 5.0 | < 0.01 | 0.98 |
| Portal phase | 18.0 ± 4.5 | 13.1 ± 4.7 | 13.5 ± 4.4 | < 0.01 | 0.86 |

Image noise: mean values ± standard deviation.

*Statistical comparison using Dunnett's test

**Table 2. Subjective assessment of image quality in the three image sets.**

| Parameter | Group A | Group B | Group C | *p value Group A vs Group C | Group B vs Group C |
|---|---|---|---|---|---|
| Sharpness | 3.4 ± 0.6 | 2.7 ± 0.5 | 3.4 ± 0.6 | 0.65 | < 0.01 |
| Noise | 2.5 ± 0.6 | 3.4 ± 0.5 | 3.2 ± 0.6 | < 0.01 | < 0.01 |
| Contrast | 2.9 ± 0.7 | 2.9 ± 0.6 | 3.1 ± 0.6 | 0.07 | 0.03 |
| Overall quality | 2.8 ± 0.6 | 3.0 ± 0.5 | 3.2 ± 0.6 | < 0.01 | 0.01 |

Data are presented as mean values ± standard deviation

*Statistical comparison using Dunnett's test

k = 0.70, 0.84, 0.83, 0.77

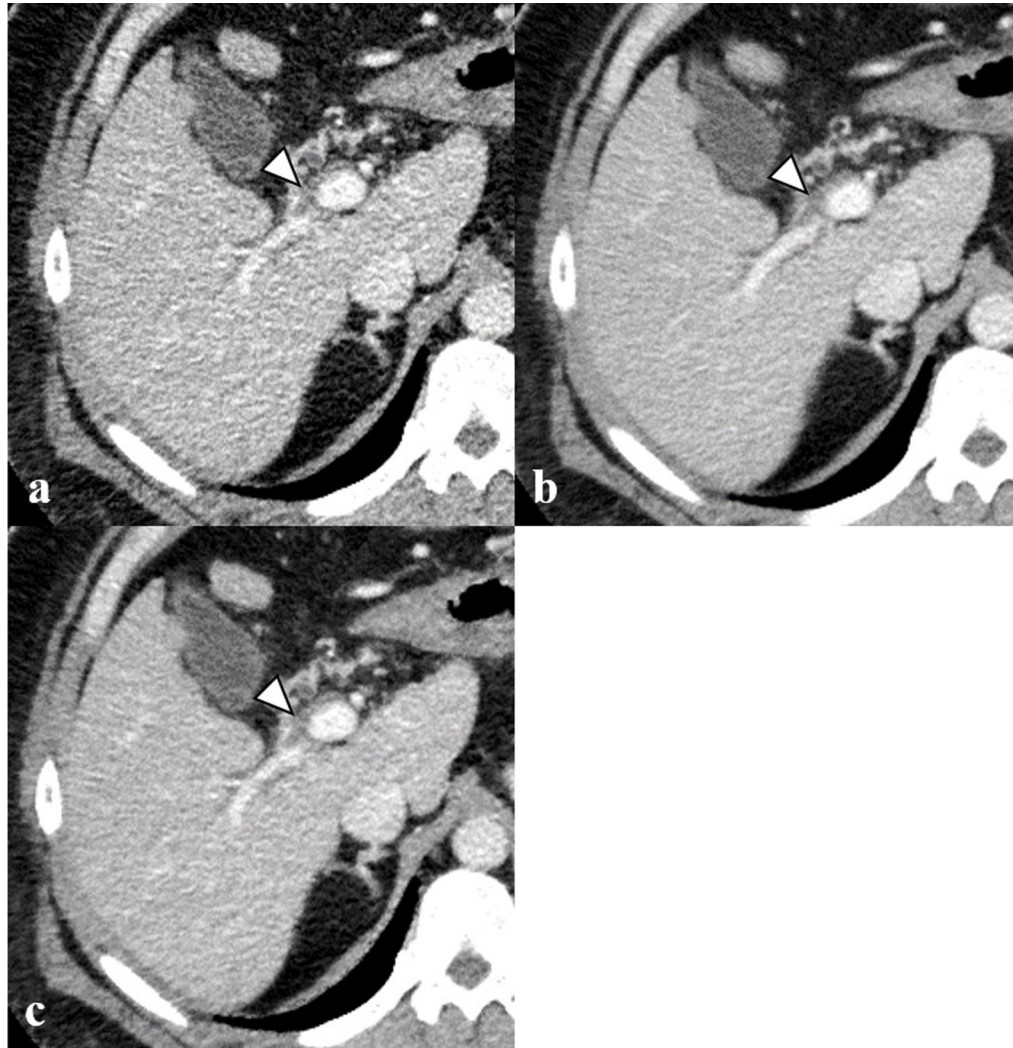

**Fig 3. Computed tomography (CT) images obtained during the portal venous phase (window width, 350 HU; window level, 40 HU) in a 43-year-old man (body weight 112 kg) with a history of liver cirrhosis.** Images of (a) group A (hybrid IR images with 1-mm slice thickness), (b) group B (hybrid IR images with 5-mm slice thickness), and (c) group C (hybrid IR images with 1-mm slice thickness denoised using iNoir) reveal that the image noise was substantially reduced in group C compared to group A. The portal vein thrombus (arrowhead) is conspicuous in (c) compared with (a) and (b).

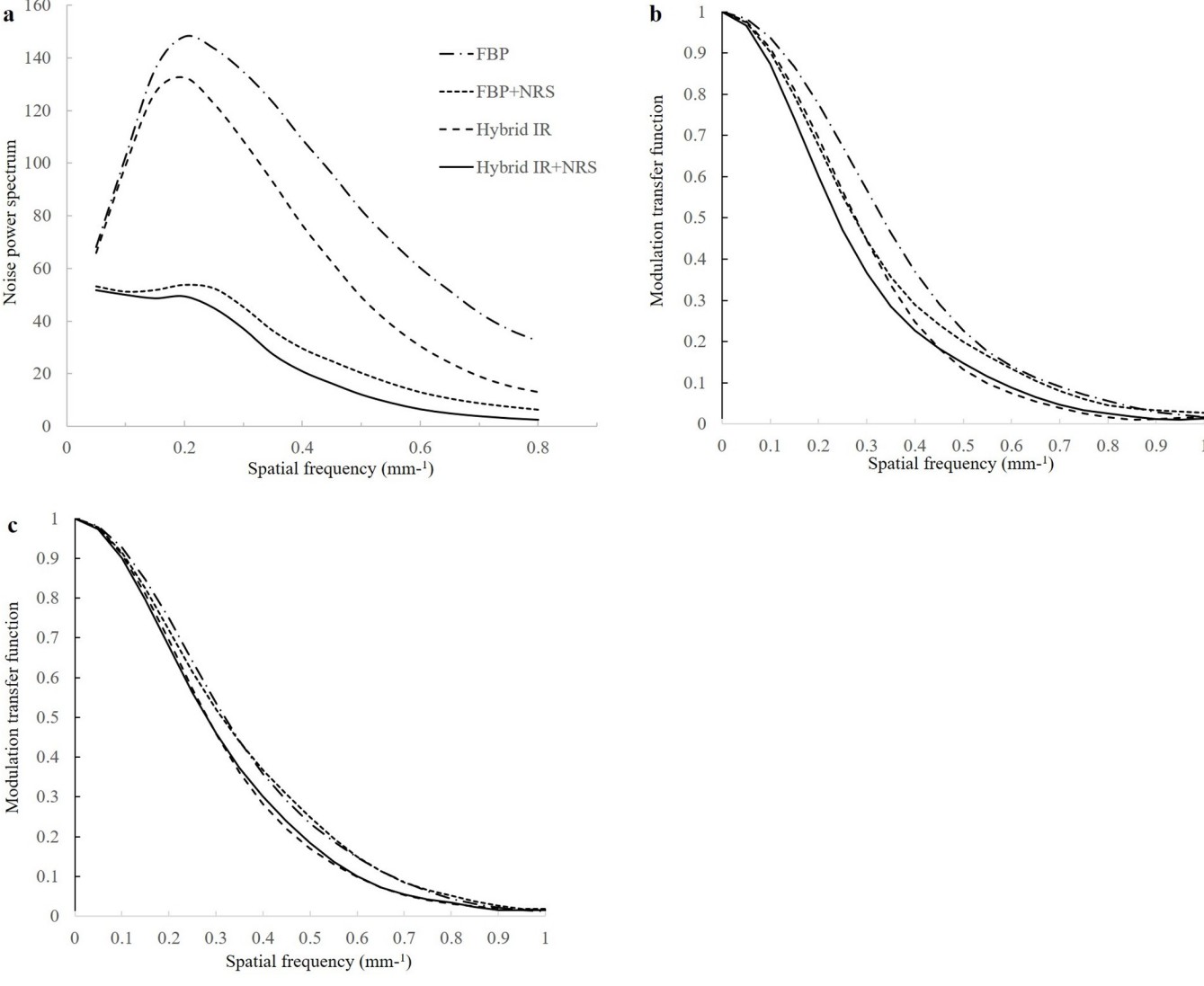

**Fig 4.** Noise power spectrum (NPS) curves (a) and modulation transfer function (MTF) curves (b:100 HU, c: 340 HU) for the four image sets: images reconstructed using filtered back projection (FBP), FBP images denoised using noise reduction software (NRS) (FBP+NRS), images reconstructed with hybrid iterative reconstruction (IR), and hybrid IR images denoised using NRS (hybrid IR+NRS).

MTF curve of all NRS protocol tended to be lower in the low frequency and tended to be equal to FBP or IR in high frequency at 100 HU (Fig 4B and 4C).

## Discussion

Thin-slice images on abdominal CT often find usage in preoperative planning, tumor classification, seeking metastases, and for emergency setting [13]. The NCCN guidelines for pancreatic cancer and cholangiocarcinoma recommend the evaluation of spread to the blood vessels and surgical planning in CT three-phase cross-sectional imaging with thinner slices for detecting tumor [2,3]. Oguro et al. reported that 22 upper gastrointestinal tract perforations and 19 lower gastrointestinal tract perforations were correctly identified as sites of perforation in 80.5% of patients when 2-mm axial and 1-mm multiplanar reconstruction (MPR) images were used [14]. In contrast, Abdelmoumene et al. compared 5-mm slice thickness with 2.5-mm slice

thickness in the diagnosis of liver metastases [15]. The results showed that 5-mm slice thickness proved to be more effective in the detection of small liver metastases. Soo et al. reported significantly improved sensitivity in 2.5-mm and 5-mm compared with 7.5-mm and 10-mm slices for the detection of liver lesions (92% in 2.5 mm, 98% in 5 mm, 78% in 7.5-mm, and 54% in 10-mm) [16]. They noticed an impaired diagnostic accuracy in 2.5-mm compared to 5-mm computed slices due to lower signal noise to ratio. We think a close relationship exists between some of these disagreement stems and background noise. It has been demonstrated that the spatial frequency distribution of noise, along with the absolute magnitude of noise, can influence image quality and ultimately object detectability [17]. In theory, slice thickness has a strong linear influence on the number of photons used to produce the images, and thinner slices use fewer photons, resulting in increased image noise and deteriorated low-contrast resolution [18], which is a critical issue because noise may substantially decrease low-contrast detectability for abdominal CT examinations. In addition, CT examinations in the overweight population are challenging because there is a tradeoff between image noise and radiation dose [6,7,12,19]. Sometimes, a modified CT protocol is adopted to obtain optimal image quality in abdominal CT, which includes high tube current and/or slow rotation time; this is associated with concerns regarding potential exposure to high-dose radiation [20]. In late 2008, GE Healthcare introduced their first hybrid ASIR algorithm for clinical use. Today, hybrid IR has been established in routine clinical practice, as it allows for substantial decrease in radiation dose through noise reduction. However, recent studies have reported that hybrid IR techniques significantly improve quantitative image quality but not low-contrast detectability in overweight patients undergoing abdominal CT [21,22].

Our study results showed that the described novel image-based denoising software "iNoir" reduced noise significantly and improved image quality for thin-slice images of abdominal CT. The NRS algorithm decreased the image noise produced by the 1-mm slice thickness by approximately 25% during the arterial phase (17.8 ± 4.5 vs 13.3 ± 5.0) and by 25% during the portal phase (18.0 ± 4.5 vs 13.5 ± 4.4) without decreasing CNR when compared with 5-mm slice thickness. In the visual evaluation, there were no statistically significant overall differences in image sharpness between hybrid IR images with 1-mm slice thickness and hybrid IR images with 1-mm slice thickness denoised using NRS. In our study, the magnitude of image noise was significantly decreased with use of the NRS algorithm, and the results of our phantom experiment differed from those obtained with hybrid IR. Although hybrid IR shows a more pronounced reduction in noise at higher spatial frequencies than at lower spatial frequencies, image noise was uniformly significantly decreased across the entire spatial frequency spectrum with the use of iNoir. We found the MTF curve, when the images reconstructed with hybrid IR and the NRS were combined, to be slightly lower at a low frequency of 100 HU. Therefore, in clinical settings, iNoir may be useful for enhanced CT or CT angiography (i.e., imaging with high contrast levels). We think this is due to the fact that, no statistically significant difference was observed in the subjective assessment of sharpness between hybrid IR images with 1-mm slice thickness and hybrid IR images with 1-mm slice thickness, denoised using NRS on portal phase CT images. Our results may support and expand on the clinical benefit of the thin-slice CT technique for abdominal CT. Future observer-based qualitative studies are warranted to investigate the effects of image-based noise reduction techniques on radiologists' subjective perception of lesion detectability, accuracy, and measurement variation at lower section thicknesses.

The iNoir takes into account the noise map from the DICOM data; it is able to selectively identify and then eliminate noise from an image on workstation. An advantage of image based denoising methods is that they are not always tied to the image reconstruction system of a CT scanner, but instead can work on images alone, potentially denoising images from several

scanners in a radiology department [23]. Several noise reduction methods have been developed to reduce radiation dose [24–27], but have generally been applied to CT images obtained using routine tube voltage settings (120 kV). With enhanced abdominal CT, noise reduction technique improves image quality at low tube voltage, high tube current, and reduced radiation dose, compared with 120 kV images [28–30]. Ideally, we would compare low-dose 80 kV denoised images to routine-dose 120 kV images. Further investigation may reveal that combined image-based noise reduction may be beneficial to low-dose abdominal images. The IRB prohibited us from performing low dose CT examinations in our routine clinical conditions simply to assess differences in image reconstruction techniques. Alternatively, a third-party vendor neutral imaging-based approach can likely be combined with low-dose imaging without compromising image quality or diagnostic confidence. This would simultaneously reduce radiation dose, retain or improve diagnostic performance, and avoid onerous increases in image noise.

Our study has several weaknesses that need to be considered. First, the BW range and the mean BW of the patients were lower than those of the overweight North American and European populations. Second, we did not evaluate the diagnostic accuracy of our technique with respect to liver tumors; rather, we focused on comparing the image quality of IR and NRS because the number of patients with liver tumors was low. Future studies are needed to evaluate the effect of this NRS on the diagnostic performance of images reconstructed using it in patients with liver tumors. Third, we only evaluated one NRS level in this study. A higher NRS level would allow greater noise reduction. However, this needs to be balanced with concerns regarding loss of image detail. Even at a 90% NRS level, our pilot study revealed that axial images had decreased image noise, but image sharpness also decreased and appeared to have an unnatural texture similarly as a limitation of iterative reconstruction techniques [27,29]. Future studies are needed to determine the optimal ratio of hybrid IR and iNoir.

In conclusion, our initial clinical results suggest that, compared with hybrid IR alone, using the novel NRS in combination with a hybrid IR could result in significant noise reduction without sacrificing image quality on thin-slice CT. This technique will likely be particularly useful for thin-slice abdominal CT in overweight patients.

## Supporting information

**S1 Table. Details of measured clinical image in the three image sets.**
(XLSX)

**S2 Table. Details of measured phantom experiments.**
(XLSX)

## Author Contributions

**Conceptualization:** Ryoichi Tanaka, Kunihiro Yoshioka.

**Data curation:** Manabu Nakayama, Misato Sone, Makoto Hamano.

**Investigation:** Yoshitaka Ota, Masayoshi Kamata, Yasuyuki Hirota.

**Writing – original draft:** Akio Tamura.

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
