## [Decision Letter · Decision Letter 0]

1 Oct 2019

PONE-D-19-22045

Feasibility of thin-slice abdominal CT in overweight patients using a vendor neutral image-based denoising algorithm: Assessment of image noise, contrast, and quality

PLOS ONE

Dear Dr. Tamura,

Thank you for submitting your manuscript to PLOS ONE. After careful consideration, we feel that it has merit but does not fully meet PLOS ONE’s publication criteria as it currently stands. Therefore, we invite you to submit a revised version of the manuscript that addresses the points raised during the review process.

We would appreciate receiving your revised manuscript by Nov 15 2019 11:59PM. To enhance the reproducibility of your results, we recommend that if applicable you deposit your laboratory protocols in protocols.io, where a protocol can be assigned its own identifier (DOI) such that it can be cited independently in the future. For instructions see: http://journals.plos.org/plosone/s/submission-guidelines#loc-laboratory-protocols

We look forward to receiving your revised manuscript.

Kind regards,

Yuchen Qiu, Ph.D.

Academic Editor

PLOS ONE

Journal Requirements:

https://link.springer.com/article/10.1007%2Fs00068-018-1021-9

In your revision ensure you cite all your sources (including your own works), and quote or rephrase any duplicated text outside the methods section. Further consideration is dependent on these concerns being addressed.

3. Thank you for including your ethics statement:

'This retrospective study was approved by our institutional review board, and the requirement for written informed consent was waived because the image data were retrospectively obtained from routine liver CT examinations.'

Additional Editor Comments (if provided):

Reviewers' comments:

Reviewer's Responses to Questions

**Comments to the Author**

1. Is the manuscript technically sound, and do the data support the conclusions?

Reviewer #1: No

Reviewer #2: Yes

2. Has the statistical analysis been performed appropriately and rigorously? 

Reviewer #1: No

Reviewer #2: Yes

3. Have the authors made all data underlying the findings in their manuscript fully available?

Reviewer #1: Yes

Reviewer #2: Yes

4. Is the manuscript presented in an intelligible fashion and written in standard English?

Reviewer #1: Yes

Reviewer #2: Yes

5. Review Comments to the Author

Reviewer #1: The authors investigated whether the novel image-based noise reduction software (NRS) improves image quality and assessed the feasibility of using this software in combination with hybrid iterative reconstruction (IR) in image quality on thin-slice abdominal CT. The results showed that compared with hybrid IR alone, the novel NRS in combination with a hybrid IR results in significant noise reduction without sacrificing image quality on CT. This study was well organized, while the significance of the result is still unclear.

Major comments:

1. The authors should state why thin-slice CT scan is important to over-weight patient.

2. The authors should show the strength and difference between novel image-based noise reduction software (NRS) and third-party vendor neutral imaging-based approach or model-based IR (MBIR) in the introduction.

3. What’s the weakness and strengthens of 1-mm CT or 5-mm CT? The authors should make it clear in the introduction.

4. The authors should contour the margins of ROIs used in this study in the figure.

5. Why the Fig 1.a and 1.c were different from gas pockets? Will the denoising process changes the image structure?

6. The authors should present the scanning result of phantom and ROI selection to demonstrate the effect of NRS.

7. The authors should add more discussion about the potential effect of this study to the clinic.

Reviewer #2: This study investigated the performance of an noise reduction method in thin-slice CT of abdomen. This topic has clinical significance and the study has proper method and reasonable statistical analysis. I have a few minor comments:

1. It would have more educational value if the authors can briefly introduce the method of NRS.

2. Discussion should also include brief comments about the reasons why NRS can achieve such performance. For example, there are criticism on some noise reduction method such as compress-sensing based noise reduction method for its over-smoothing and patchy appearance. How is the NRS different from this method to maintain image sharpness?

3. The noise power spectrum shows high frequency noise is suppressed by NRS. It may cause the loss of high frequency spatial information and then lower the spatial resolution. However, this study lacks a quantitative evaluation on the change of spatial resolution by NRS. Although the subjective study includes the score on sharpness, it can not serve as a benchmark for other studies. I recommend a phantom study with measurement on line pairs on the three groups.

6. PLOS authors have the option to publish the peer review history of their article (what does this mean?). If published, this will include your full peer review and any attached files.

Reviewer #1: No

Reviewer #2: No

---

## [Author Response · Author response to Decision Letter 0]

22 Oct 2019

We appreciate the reviewers’ insightful comments, which have helped us significantly improve our paper. Below are our point-by-point responses to the reviewers’ comments. We hope that our revised manuscript is now acceptable for publication.

RESPONSES TO REVIEWER 1: 

Comment 1: 

The authors should state why thin-slice CT scan is important to over-weight patient.

Response:

Based on the reviewer’s comment, we have added the following sentences to the Introduction.

“Recent advancements in MDCT have allowed image acquisition with thinner collimation and more rapid scan times, thus, enabling better image resolution to delineate abdominal disease and abnormalities. In most clinical practice guidelines and recommendations for patients with abdominal cancers, such as pancreatic cancer and cholangiocarcinoma, the CT technique involves multiphase thin-section image acquisition to optimize the evaluation of spread to the blood vessels [2-4]. The thinner layer provides better detail and spatial resolution; conversely, noise in CT image increases with a thinner slice [5].”

Comment 2: 

The authors should show the strength and difference between novel image-based noise reduction software (NRS) and third-party vendor neutral imaging-based approach or model-based IR (MBIR) in the introduction.

Response:

Based on the reviewer’s comment, we have added the following sentences to the Discussion.

“The iNoir takes into account the noise map from the DICOM data; it is able to selectively identify and then eliminate noise from an image on workstation. An advantage of image based denoising methods is that they are not always tied to the image reconstruction system of a CT scanner, but instead can work on images alone, potentially denoising images from several scanners in a radiology department.”

Comment 3: 

What’s the weakness and strengthens of 1-mm CT or 5-mm CT? The authors should make it clear in the introduction.

Response:

Based on the reviewer’s comment, we have added the following sentences to the Introduction.

“CT image noise strongly depends on the patient's body size and the tube current applied during data acquisition. It is theoretically a priori possible to increase the number of photons and, thus, increase the radiation dose, and to obtain image quality even with a 1-mm slice compared to a 5-mm slice acquired with a lower tube current [8].”

We have added the following sentences to the Discussion.

“NCCN guidelines for pancreatic cancer and cholangiocarcinoma recommend the evaluation of spread to the blood vessels and surgical planning in CT three-phase cross-sectional imaging with thinner slices for detecting tumor [2,3]. Oguro et al. reported that 22 upper gastrointestinal tract perforations and 19 lower gastrointestinal tract perforations were correctly identified as sites of perforation in 80.5% of patients when 2-mm axial and 1-mm multiplanar reconstruction (MPR) images were used [15].”

“We think a close relationship exists between some of these disagreement stems and background noise. It has been demonstrated that the spatial frequency distribution of noise, along with the absolute magnitude of noise, can influence image quality and ultimately object detectability [18].”

Comment 4: 

The authors should contour the margins of ROIs used in this study in the figure.

Response:

Based on the reviewer’s comment, we have added Figure 1 in Materials and methods.

Comment 5: 

Why the Fig 1.a and 1.c were different from gas pockets? Will the denoising process changes the image structure?

Response:

Based on the reviewer’s comment, we have replaced the figure.

Comment 6: 

The authors should present the scanning result of phantom and ROI selection to demonstrate the effect of NRS.

Response:

Based on the reviewer’s comment, we have added CT images showing the 220-mm diameter cylindrical water phantom and ROI selection as Figure 2.

Comment 7: 

The authors should add more discussion about the potential effect of this study to the clinic.

Response:

Based on the reviewer’s comment, we have added the following sentences to the Discussion.

“Several noise reduction methods have been developed to reduce radiation dose [24-27], but have generally been applied to CT images obtained using routine tube voltage settings (120 kV). With enhanced abdominal CT, noise reduction technique improves image quality at low tube voltage, high tube current, and reduced radiation dose, compared with 120 kV images [28-30]. Ideally, we would compare low-dose 80 kV denoised images to routine-dose 120 kV images. Further investigation may reveal that combined image-based noise reduction may be beneficial to low-dose abdominal images. The IRB prohibited us from performing low dose CT examinations in our routine clinical conditions simply to assess differences in image reconstruction techniques. Alternatively, a third-party vendor neutral imaging-based approach can likely be combined with low-dose imaging without compromising image quality or diagnostic confidence. This would simultaneously reduce radiation dose, retain or improve diagnostic performance, and avoid onerous increases in image noise.”

RESPONSES TO REVIEWER 2: 

Comment 1: 

It would have more educational value if the authors can briefly introduce the method of NRS.

Response to Comment 1

Based on the reviewer’s comment, we have added the following sentences to the Discussion.

“The iNoir takes into account the noise map from the DICOM data; it is able to selectively identify and then eliminate noise from an image on workstation. An advantage of image-based denoising methods is that they are not always tied to the image reconstruction system of a CT scanner, but instead can work on images alone, potentially denoising images from several scanners in a radiology department”

They incorporate company know-how into their algorithms, which are protected as trade secrets.

Comment 2: 

Discussion should also include brief comments about the reasons why NRS can achieve such performance. For example, there are criticism on some noise reduction method such as compress-sensing based noise reduction method for its over-smoothing and patchy appearance. How is the NRS different from this method to maintain image sharpness?

Comment 3: 

The noise power spectrum shows high frequency noise is suppressed by NRS. It may cause the loss of high frequency spatial information and then lower the spatial resolution. However, this study lacks a quantitative evaluation on the change of spatial resolution by NRS. Although the subjective study includes the score on sharpness, it can not serve as a benchmark for other studies. I recommend a phantom study with measurement on line pairs on the three groups.

Response to Comment 2&3:

Based on the reviewer’s comment, we conducted additional phantom experiments to evaluate the modulation transfer function (MTF). We have added the MTF data and Figure 4 in Results.

Based on the reviewer’s comment, we have added the following sentences to the Discussion.

“We found the MTF curve, when the images reconstructed with hybrid IR and the NRS were combined, to be slightly lower at a low frequency of 100 HU. Therefore, in clinical settings, iNoir may be useful for enhanced CT or CT angiography (i.e., imaging with high contrast levels). We think this is due to the fact that, no statistically significant difference was observed in the subjective assessment of sharpness between hybrid IR images with 1-mm slice thickness and hybrid IR images with 1-mm slice thickness denoised using NRS on portal phase CT images.”

---

## [Decision Letter · Decision Letter 1]

2 Dec 2019

Feasibility of thin-slice abdominal CT in overweight patients using a vendor neutral image-based denoising algorithm: Assessment of image noise, contrast, and quality

PONE-D-19-22045R1

Dear Dr. Tamura,

We are pleased to inform you that your manuscript has been judged scientifically suitable for publication and will be formally accepted for publication once it complies with all outstanding technical requirements.

With kind regards,

Yuchen Qiu, Ph.D.

Academic Editor

PLOS ONE

Additional Editor Comments (optional):

Reviewers' comments:

Reviewer's Responses to Questions

**Comments to the Author**

1. If the authors have adequately addressed your comments raised in a previous round of review and you feel that this manuscript is now acceptable for publication, you may indicate that here to bypass the “Comments to the Author” section, enter your conflict of interest statement in the “Confidential to Editor” section, and submit your "Accept" recommendation.

Reviewer #1: All comments have been addressed

Reviewer #2: All comments have been addressed

2. Is the manuscript technically sound, and do the data support the conclusions?

Reviewer #1: Yes

Reviewer #2: Yes

3. Has the statistical analysis been performed appropriately and rigorously? 

Reviewer #1: Yes

Reviewer #2: Yes

4. Have the authors made all data underlying the findings in their manuscript fully available?

Reviewer #1: Yes

Reviewer #2: Yes

5. Is the manuscript presented in an intelligible fashion and written in standard English?

Reviewer #1: Yes

Reviewer #2: Yes

6. Review Comments to the Author

Reviewer #1: I appreciate the nice work from the authors and agree with the publication. The authors have pereformed serious revision to address my comments.

Reviewer #2: The phantom results improve the completeness of the study and useful for comparison with other studies. I do not have further comments.

7. PLOS authors have the option to publish the peer review history of their article (what does this mean?). If published, this will include your full peer review and any attached files.

Reviewer #1: Yes: TIANYE NIU

Reviewer #2: No

---

## [Editor Report · Acceptance letter]

9 Dec 2019

PONE-D-19-22045R1 

Feasibility of thin-slice abdominal CT in overweight patients using a vendor neutral image-based denoising algorithm: Assessment of image noise, contrast, and quality 

Dear Dr. Tamura:

I am pleased to inform you that your manuscript has been deemed suitable for publication in PLOS ONE. Congratulations! Your manuscript is now with our production department. 

With kind regards,

on behalf of

Dr. Yuchen Qiu 

Academic Editor

PLOS ONE